# Comparative Use of Contralateral and Sham-Operated Controls Reveals Traces of a Bilateral Genetic Response in the Rat Brain after Focal Stroke

**DOI:** 10.3390/ijms23137308

**Published:** 2022-06-30

**Authors:** Ivan B. Filippenkov, Julia A. Remizova, Alina E. Denisova, Vasily V. Stavchansky, Ksenia D. Golovina, Leonid V. Gubsky, Svetlana A. Limborska, Lyudmila V. Dergunova

**Affiliations:** 1Department of Molecular Bases of Human Genetics, Institute of Molecular Genetics of National Research Center “Kurchatov Institute”, Kurchatov Sq. 2, 123182 Moscow, Russia; utoshkautoshka@gmail.com (J.A.R.); bacbac@yandex.ru (V.V.S.); zefeni@yandex.ru (K.D.G.); limbor@img.msk.ru (S.A.L.); lvd@img.msk.ru (L.V.D.); 2Department of Neurology, Neurosurgery and Medical Genetics, Pirogov Russian National Research Medical University, Ostrovitianov Str. 1, 117997 Moscow, Russia; dalina543@gmail.com (A.E.D.); gubskii@mail.ru (L.V.G.); 3Federal Center for the Brain and Neurotechnologies, Federal Biomedical Agency, Ostrovitianov Str. 1, Building 10, 117997 Moscow, Russia

**Keywords:** tMCAO, RNA-Seq, contralateral hemisphere of the rat brain, sham-operated rats, gene expression, transhemispheric differences

## Abstract

Ischemic stroke is a multifactorial disease with a complex etiology and global consequences. Model animals are widely used in stroke studies. Various controls, either brain samples from sham-operated (SO) animals or symmetrically located brain samples from the opposite (contralateral) hemisphere (CH), are often used to analyze the processes in the damaged (ipsilateral) hemisphere (IH) after focal stroke. However, previously, it was shown that focal ischemia can lead to metabolic and transcriptomic changes not only in the IH but also in the CH. Here, using a transient middle cerebral artery occlusion (tMCAO) model and genome-wide RNA sequencing, we identified 1941 overlapping differentially expressed genes (DEGs) with a cutoff value >1.5 and *Padj* < 0.05 that reflected the general transcriptome response of IH subcortical cells at 24 h after tMCAO using both SO and CH controls. Concomitantly, 861 genes were differentially expressed in IH vs. SO, whereas they were not vs. the CH control. Furthermore, they were associated with apoptosis, the cell cycle, and neurotransmitter responses. In turn, we identified 221 DEGs in IH vs. CH, which were non-DEGs vs. the SO control. Moreover, they were predominantly associated with immune-related response. We believe that both sets of non-overlapping genes recorded transcriptome changes in IH cells associated with transhemispheric differences after focal cerebral ischemia. Thus, the specific response of the CH transcriptome should be considered when using it as a control in studies of target brain regions in diseases that induce a global bilateral genetic response, such as stroke.

## 1. Introduction

Ischemic stroke is the most common cause of death and disability in the population of developed countries [1,2]. A recent economic analysis performed in Europe reported that stroke cost the area EUR 60 billion, with health care accounting for EUR 27 billion (45%), representing 1.7% of health expenditures in the area [3]. The genomic component occupies a significant place in the pathogenesis, diagnosis, and treatment of ischemic stroke. A wide range of high-throughput genomic and post-genomic technologies, as well as the development of model systems, have allowed the generation of data that afford a better understanding of effective approaches to overcome challenges in stroke. Previously, a multiple transcriptomic response in the damaged (ipsilateral) hemisphere (IH) of the brain was revealed in the ischemic condition in model animals [4,5,6,7,8,9,10,11,12]. The rodent ischemia model systems of permanent middle cerebral artery occlusion (pMCAO) and transient middle cerebral artery occlusion (tMCAO) were used in these studies. Reverse transcription polymerase chain reaction (RT-PCR) in real-time, microarray, and genome-wide RNA sequencing (RNA-Seq) were used to detect the transcriptomic response. Thus, different studies have succeeded in identifying the modulation of the expression pattern of genes involved in metabolic cell activity, neurotransmission, inflammation, immune response, apoptosis, and stress response in the IH. Moreover, previously, we reported the brain expression profile of key proteins involved in inflammation and cell death processes (MMP-9, c-Fos, and JNK), as well as in neuroprotection and recovery (CREB), as detected at 24 h after tMCAO in the IH [13].

It should be noted that samples from the contralateral hemisphere (CH) are often used as a control to study the processes occurring in the IH of the brain after focal stroke [14,15]. However, previously, it was shown that focal ischemia can lead to metabolic changes not only in the damaged IH but also in the CH of the brain [16,17]. Abe et al. termed this phenomenon transhemispheric diaschisis (TD) [17]. Diaschisis is defined as a functional inhibition of the brain distant from the original site of injury [18]. It is known that spreading depression/depolarization (SD) also plays a significant role in the pathophysiology of various diseases, including ischemic stroke [19,20,21,22]. SD is a temporary wave of near-complete depolarization of neurons and glia that is associated with massive transmembrane ion and water shifts caused by massive damage [19,23]. It has been shown that TD caused by cortical SD is associated with impaired glucose and oxygen metabolism, as well as reduced cerebral blood flow, in adjacent areas of the brain [24]. More than 10 years ago, it was shown that gene expression patterns can change in the CH after focal cerebral ischemia. In fact, during ischemia, the expression levels of neurotrophin-encoding genes were significantly changed in the brain regions of the CH compared with the sham-operated (SO) control [25,26,27]. Recently, genome-wide mRNA profiling in the CH of the mouse brain after tMCAO was attempted for the first time [28,29]. Incidentally, the set of differentially expressed genes (DEGs) in the CH only partially overlapped with the set of DEGs in the IH [29]. These results indicate the complex process of formation of cellular responses in the CH during ischemia. To study gene expression changes in the IH, the use of appropriate samples of SO animals is a rational alternative to using the CH as a control [7,30,31].

In the present study, a tMCAO (90 min) rat model was implemented using an MRI control. An RNA-Seq analysis of IH subcortical brain structures was carried out using SO as well as CH control brain samples. PCR verification of the changes in the expression of individual genes in the IH vs. SO and CH controls was performed. As a result, both types of controls allowed the identification of overlapping DEGs that reflected the general transcriptome response of IH subcortical cells at 24 h after tMCAO. Concomitantly, we identified sets of non-overlapping genes that were unique for use in SO and CH controls. Such DEGs can record the transcriptome changes in IH subcortical cells associated with transhemispheric differences under ischemia model conditions. The functional annotation of gene sets was performed using an enrichment analysis. Thus, the comparative use of contralateral and SO controls revealed traces of a bilateral genetic response in the brain after focal stroke. 

## 2. Results

### 2.1. MRI-Based Detection of the Location and Volume of Ischemic Foci

According to the MRI data, we detected the location and volume of ischemic foci in animals after tMCAO. A typical example of diffusion-weighted imaging (DWI) with an apparent diffusion coefficient (ADC) map and T2-weighted imaging (T2 WI) scans of the formation of the ischemic injury in the brain of rats at 24 h after tMCAO are shown in Figure 1. The hyperintense signal corresponded to ischemic damage in the territory of the right middle cerebral artery in all test rats. Thus, all rats in the IR group had a focal hemispheric lesion lying in the ipsilateral subcortex region that spread to the adjacent cortex. Concomitantly, no pathological changes in the CH of ischemic rats were observed, according to the MRI data.

Appendix A shows the T2 WI data pertaining to the size (in mm^3^) of the ischemic brain damage in rats in the IR group.

### 2.2. RNA-Seq Analysis of the Effect of IR on the Level of mRNAs in Subcortical Structures of the IH Related to the CH

The use of RNA-Seq allowed the evaluation of the transcriptional activity of 17,367 genes (mRNAs) in the subcortical structures of the rat brain at 24 h after tMCAO. We used two types of control groups to study the transcriptomic effects in the IH, namely symmetrically located areas in the CH and corresponding brain samples of SO rats (Figure 2a).

First, we used the CH samples as a control of the IH in the IR-i vs. IR-c comparison, which led to the identification of 2162 DEGs (1032 up- and 1130 downregulated) (Appendix A). A volcano plot illustrates the differences in mRNA expression between the IR-i and IR-c groups (Figure 2b). The top 10 DEGs included *Il1rn*, *Serpina3n*, *Itgad*, *Il6*, and *Cxcl1*, which were upregulated by more than 26-fold, whereas the *Ltc4s*, *Wnk3*, *Erbb4*, *Dok6*, and *Acvr1c* DEGs were downregulated in this comparison group by more than seven-fold (Figure 2c).

### 2.3. RNA-Seq Analysis of the Effect of IR on the Level of mRNAs in Subcortical Structures of the IH Related to the SO Rats

We next assessed the effect of focal IR on the mRNA levels of genes in subcortical structures of the IH relative to the corresponding SO control (Figure 2a). We identified 2802 DEGs with 1390 upregulated and 1412 downregulated mRNAs in the rat ipsilateral subcortex (IR-i vs. SO-r) (Appendix A). The volcano plot illustrates the differences in mRNA expression between the IR-i and SO-r groups (Figure 2d). The top five upregulated DEGs *Hspa1a*, *Ptx3*, *H19*, *Atf3*, and *Il11* exhibited an increase of ≥63-fold in their expression in the subcortex of the IH at 24 h after tMCAO. Concomitantly, the top five downregulated DEGs *Dok6*, *Erbb4*, *Wnk3*, *Acvr1c*, and *Kcnh5* exhibited a decrease of ≥7-fold in their mRNA levels in IR-i vs. SO-r (Figure 2e).

### 2.4. Comparisons of the Results of DEGs in Subcortical Structures of the IH at 24 h after tMCAO, as Assessed Using CH and SO Controls

A violin plot was utilized to compare the distributions of the intensities of the expression changes in DEGs from each pairwise comparison of IR-i vs. SO-r and IR-i vs. IR-c (Figure 3a). Moreover, we identified 1941 overlapping DEGs using two controls in a pairwise comparison of IR-i vs. IR-c and IR-i vs. SO-r (Figure 3b). All of them codirectionally changed the mRNA level in both comparison groups (Appendix A), with the exception of the *Parpbp* gene, which encodes the PARP1 binding protein and was downregulated in relation to the sham operation but upregulated in relation to the CH of ischemic rats. A hierarchical cluster analysis of all DEGs in IR-i vs. SO-r and IR-i vs. IR-c illustrated the mostly codirectional changes in gene expression in IH using both CH and SO controls (Figure 3c).

Furthermore, the Venn diagram shows that 861 DEGs were unique for IR-i vs. SO-r (Figure 3b, Appendix A). For example, the expression of the *Cxcr2, Apold1, Clk2, Egr2, Cdkn2c, Kcne2, Sostdc1, Kcnj13, Tnnt2,* and *Slco1a5* genes, among others, was significantly altered in IR-i vs. SO-r (using a SO control), whereas they were non-DEGs in IR-i vs. IR-c (using contralateral samples from the “IR-c” group as the control) (Figure 3d). Thus, these genes may be considered “lost results” when using the CH as a control. Concomitantly, there were 221 such DEGs for the IR-i vs. IR-s comparison in the Venn diagram (Figure 3b). For example, the *Cxcl1, Lif, S100a9, Tmem202, Slc47a1, Abcg2, Sult5a1, Ptprv, Pla2g3, Tnnc1,* and other genes were DEGs in IR-i vs. IR-c, but non-DEGs in IR-i vs. SO-r (Figure 3e, Appendix A). These genes may be considered “redundant results” when characterizing the transcriptional activity of cells in the IH after stroke.

The transcriptome analysis performed as a comparison between the contralateral ischemic hemisphere and the sham-operated control left hemisphere (IR-c vs. SO-l) identified 164 DEGs. The comparison of these DEGs with the results obtained using two controls in pairwise analyses of IR-i vs. SO-r and IR-i vs. IR-c (Figure 3b) allowed us to identify four sets of genes, which were illustrated in a Venn diagram (Appendix A). First, 86 genes were altered by all pairwise comparisons and overlapped with 1941 genes that were identified using SO and CH controls. Second, 44 genes overlapped with sets of “lost results”. All of them (e.g., *Apold1, Clk2, Egr2, Sostdc1, Kcnj13, Slco1a5*) changed their expression codirectionally in both the IR-i vs. SO-r and IR-c vs. SO-l pairwise comparisons; however, such genes were non-DEGs in IR-i vs. IR-c (Figure 4a). Third, six DEGs overlapped with sets of “redundant results”; the *S100a9* and *Plagl1* genes were upregulated in IR-i vs. IR-c but downregulated in IR-c vs. SO-l. Conversely, the *Arc, Abcg2, Pou3f1,* and *Pla2g3* genes were downregulated in IR-i vs. IR-c but upregulated in IR-c vs. SO-l. Moreover, all of these genes were non-DEGs in IR-i vs. SO-r (Figure 4b). Fourth, 28 genes were DEGs in IR-c vs. SO-l exclusively (Appendix A). As a result, the significant expression changes of genes in IR-c vs. SO-l reflect post-ischemic activity in CH and demonstrate the problems of using CH as a control.

### 2.5. Real-Time Reverse Transcription Polymerase Chain Reaction (RT-PCR) Verification of the RNA-Seq Results of the Subcortical Structures of the IH at 24 h after tMCAO, as Assessed Using CH and SO Controls

An RT-PCR analysis of the expression of nine genes (*Hspb1, Mmp9, Socs3, Fos, Cp, Cyr61, Bcl2l2, Cartpt,* and *Pla2g3*) was used to verify the RNA-Seq results in two pairwise comparisons, i.e., IR-i vs. SO-r and IR-i vs. IR-c. The primers used are shown in Appendix A. The *Hspb1, Mmp9, Socs3, Fos,* and *Cyr61* genes were DEGs in both IR-i vs. SO-r and IR-i vs. IR-c. The *Cp* gene was a DEG in IR-i vs. SO-r, but a non-DEG in IR-i vs. IR-c; whereas the *Pla2g3* gene was a DEG in IR-i vs. IR-c, but a non-DEG in IR-i vs. SO-r. Moreover, the *Bcl2l2* and *Cartpt* genes were non-DEGs in both IR-i vs. SO-r and IR-i vs. IR-c pairwise comparisons. The real-time RT-PCR results adequately confirmed the RNA-Seq data (Figure 5).

### 2.6. Signaling Pathways Associated with DEGs in the Two Rat Brain Hemispheres at 24 h after tMCAO

DAVID, which is a pathway-enrichment analysis, was used to annotate the signaling pathways associated with DEGs in the rat brain subcortex under tMCAO model conditions in pairwise comparisons of IR-i vs. SO-r and IR-i vs. IR-c. We used the KEGG PATHWAY (KP), REACTOME PATHWAY (RP), and WIKIPATHWAYS (WP) signaling pathway annotations for DEGs from the two following pairwise comparisons: IR-i vs. SO-r and IR-i vs. IR-c (Appendix A). We found 171 KP (Figure 6a), 44 RP (Appendix A), and 12 WP (Appendix A) annotations for the comparison groups in total.

The Venn diagram presented in Figure 6a shows 143 overlapped signaling pathways that were annotated according to the most abundant KP annotations between IR-i vs. SO-r and IR-i vs. IR-c. The top five signaling pathways of KP and corresponding *Padj*-values as well as the number of up- and downregulated DEGs in IR-i vs. SO-r and IR-i vs. IR-c among the overlapped annotations are presented in Figure 6b and c, respectively. Therefore, IR activated the expression of genes related to the mitogen-activated protein kinase (MAPK) signaling pathway, proteoglycans in cancer, lipid and atherosclerosis, Rap1 signaling pathway, etc., and suppressed the expression of genes related to the neuronal system of glutamatergic synapse, oxytocin, calcium signaling pathways, etc., in IH, regardless of the type of control used (SO-r or IR-c).

The Venn diagram depicted in Figure 6a shows annotations that lie outside the intersection and were unique for the effect of IR in IH transcriptome vs. the SO and CH controls. The characteristics of the most significant signaling pathways (with a minimal *Padj*) that were unique for IR-i vs. SO-r and were annotated by KP are presented in Figure 6d, whereas similar data for IR-i vs. IR-c are presented in Figure 6e. We identified apoptosis-multiple species, transforming growth factor (TGF)-beta, protein processing in the endoplasmic reticulum, and other signaling pathways that were unique for IR-i vs. SO-r and associated with upregulated DEGs predominantly. Moreover, there were endocrine and other factor-regulated calcium reabsorption, synaptic vesicle cycle, insulin, and other signaling pathways that were also unique for IR-i vs. SO-r and associated with downregulated DEGs predominantly (Figure 6d, Appendix A). Concomitantly, we found a powerful immune-related genetic response that was specific for IR-i vs. IR-c. Namely, hematopoietic cell lineage, antigen processing and presentation, antifolate resistance, inflammatory bowel disease, and other signaling pathways were predominantly associated with upregulated DEGs in IR-i vs. IR-c.

The KEGG enrichment results were mostly reproduced using RP and WP annotations, including the presentation of neuronal systems (RP), MAPK (WP), signaling by interleukins (RP), IL-3 (WP), IL-5 (WP), and other signaling pathways. The characteristics of the top five most significant RP and WP annotations that were common or unique between IR-i vs. SO-r and IR-i vs. IR-c pairwise comparisons are shown in Appendix A, respectively. It should be noted that only one signaling pathway (Phase 0—rapid depolarization, RP) was unique for IR-i vs. IR-c and associated with downregulated DEGs predominantly. Thus, it was identified using WP annotation and included in the top five most significant signaling pathways (Appendix A).

## 3. Discussion

In this study, we studied the role of the genetic factor of stroke that may underlie the observed post-ischemic events in the brain away from the source of injury. Here, a tMCAO (90 min) rat model that reflected ischemic stroke events in humans [32,33] was implemented under MRI control. An RNA-Seq analysis of the IH subcortical brain structures was carried out using two different controls, namely symmetrically located CH samples and corresponding brain samples of the right hemisphere of an additional group of SO rats. When we profiled the transcriptional activity of 17,367 genes (mRNAs) in the subcortical structures of the rat brain at 24 h after tMCAO, 1941 overlapping DEGs were revealed using SO and CH controls. The overlapped DEGs most likely reflected the general transcriptome response of IH subcortical cells at 24 h after tMCAO. Most of them exhibited codirectional gene expression changes in the IH using both CH and SO controls. These were the genes that were retained as true results in the differential expression analysis, including those obtained using the CH controls. Among them were the *Hspb1, Mmp9, Socs3, Fos,* and *Cyr61* genes. Such genes were verified by us with high confidence using both RNA-Seq and real-time RT-PCR methods on extended sample groups. A functional enrichment analysis showed a significant association between transcriptome changes detected after ischemia and the modulation of the activity of inflammation systems, the immune response, as well as the neurosignaling and neuroreception systems in the rat brain. These effects were overlapped when using both the CH and SO controls. In particular, genes that were upregulated were predominantly involved in the presentation of inflammatory and immune signaling systems, whereas downregulated genes were predominantly involved in the presentation of neurotransmitter systems. The data obtained here largely reproduced previously established effects after ischemia in the IH [7,29].

However, the gene expression profile overlapped only partially when using both controls. Therefore, non-overlapped DEGs included two groups of results, in reality. The first group included 861 DEGs that were obtained using the SO control exclusively. Thus, these genes may be considered “lost results” when using the CH as a control. One of these genes, the *Cp* gene, encodes the ferroxidase enzyme ceruloplasmin, which notably attracted attention in this case. Therefore, differential changes in the expression of this gene were verified by us using both genome-wide sequencing and local PCR techniques. This gene was a DEG only when using the SO control. Consequently, *Cp* is among the genes that remain at risk of being overlooked when studying the transcriptome of the IH after stroke using CH controls. It should be noted that *Cp* may play a significant role in the functioning of damaged cells. Thus, a correlation between brain ceruloplasmin expression after experimental intracerebral hemorrhage and protection against iron-induced brain injury was revealed [34]. Furthermore, differential effects of and changes in ceruloplasmin in the hippocampal CA1 region between adult and aged gerbils after transient cerebral ischemia were observed [35]. Moreover, a genetic association between ceruloplasmin and cardiovascular risk was noted [36,37,38].

Concomitantly, 221 non-overlapping DEGs were obtained using the CH control exclusively. Such genes may be considered “redundant results” when characterizing the transcriptional activity of cells in the IH after stroke. The *Pla2g3* gene, which encodes a group 3 secretory phospholipase A2 precursor, is an example of genes representing “redundant results”. In other words, the *Pla2g3* gene was a DEG only when using the CH control and was discovered by us using both RNA-Seq and real-time RT-PCR methods.

It should be noted that the presence of “lost results” and “redundant results” may not only reflect the insufficient quality of the CH as a control for studying the consequences of focal stroke; but also, it may indicate the presence of a complex regulation of cellular responses in the brain away from the ischemic focus. Identification of DEGs in comparison between the CH versus the SO control of left hemisphere additionally proves such thesis. The genes that were in the set of non-overlapped DEGs identified here are more prone to dynamic changes in expression. It was shown that the phospholipase A2 activity induced by Pla-family genes was associated not only with ischemic stroke but also with Alzheimer’s disease and vascular dementia [39]. Moreover, the *Pla2g3* gene was included in the top five upregulated genes in the rat hippocampus in response to acute stress [40]. Previous studies have shown that the anti-inflammatory S-2474 drug rescues cortical neurons from human group IIA secretory phospholipase A(2)-induced apoptosis [41]. However, chronic intracerebroventricular delivery of the secretory phospholipase A2 inhibitor 12-epi-scalaradial does not improve the outcome after focal cerebral ischemia–reperfusion in rats [42]. Moreover, previously, we showed that the *Pla2g3* gene was upregulated by 2.6-fold in the subcortical structures of the IH at 24 h after tMCAO relative to the corresponding SO control [7]. In that study, we used long-term anesthesia, whereas here the anesthesia duration was much shorter. As a result, the *Pla2g3* mRNA level was changed by <1.5-fold and was classified as a non-DEG in IH vs. SO in this study. Thus, numerous factors can affect the expression of genes such as *Pla2g3*. Among them are both essential pathogenetic processes of various brain diseases or factors of neuroprotective effects of drugs, as well as more subtle factors of localization of a particular brain sample or the presence/absence of anesthesia in the experiment, among others. It is possible that genes included in the set of non-overlapped DEGs obtained by us using SO and CH controls exclusively are objects of multiple (pleiotropic) regulations and should be given increased attention.

In our data, the extensive inflammatory response drew special attention as a result of the functional enrichment analysis of DEGs using three annotation databases. However, the response was heterogeneous when we used the CH or SO control. One part of the data, including MAPK, TNF, NF-kappa B, immune system, and other inflammatory signaling pathways, overlapped when using both CH and SO controls. Concomitantly, a significant portion of inflammatory pathways, including antigen processing and presentation, JAK-STAT, cytokine signaling in the immune system, IL-3, IL-5, IL-17, and other signaling pathways, were detected using only CH controls. It should be noted that stroke causes a rapid and massive infiltration of peripheral immune cells [43,44]. Moreover, it has been demonstrated that the infiltration of peripheral immune cells occurs in areas remote from the sites of the primary injury during the progression of injury and brain repair [29,44]. Thus, the cumulative set of inflammatory effects may exceed the real one in IH when using CH as a control. Conversely, the results indicate an expansion of the zone of inflammatory and immune reactions into the zone of the CH in response to focal ischemia–reperfusion injury. It is possible that corresponding effects will also be observed at the protein level. However, the transcriptome profile does not provide reliable criteria for choosing individual proteins for analysis to show the difference in the use of the SO and CH controls at the protein level. The correlation between transcriptome and proteome profiles is non-linear because of post-transcriptional, translational, and post-translational regulatory events [45,46,47,48]. We believe that further integrative functional genetic and proteomic analyses of the processes occurring in CH will allow the establishment of specific regulatory axes associated with the spread of the effects of focal ischemia to distanced areas of the brain. This was one of the limitations of this study, which will be overcome in the future.

Moreover, one of the problems for clinical trials is related to the correlation between the preclinical results obtained using animal models and human parameters [49,50,51]. It cannot be ruled out that the quality of clinical uses is limited by problem of control samples at the preclinical stage. Here, we characterize genes that can produce “redundant” or “lost” results of differential expression when using the contralateral hemisphere as a control under tMCAO conditions. We believe that the correct choice of controls will contribute to a more reliable interpretation of the data obtained using animal models of experimental ischemia at the preclinical stage. Thus, it is possible to achieve more effective diagnostic and therapeutic approaches to the treatment of ischemic stroke at the clinical stage.

## 4. Materials and Methods

### 4.1. Animals

White 2-month-old male rats of the Wistar line (weight, 200–250 g) were obtained from the AlCondi Ltd animal breeding house, Moscow, Russia. The animals were divided into the “sham operation” (SO) and “ischemia–reperfusion” (IR) groups.

### 4.2. Transient Middle Cerebral Artery Occlusion (tMCAO) Model in Rats 

The tMCAO rat model was induced via endovascular occlusion of the right middle cerebral artery using a monofilament (Doccol Corporation, Sharon, MA, USA) for 90 min using the method of Koizumi et al. [52] with modifications. The tMCAO details are described in Appendix A. Before the surgical procedure, rats were anesthetized using 3% isoflurane. The anesthesia was maintained using 1.5%–2% isoflurane and the EZ-7000 small animal anesthesia system (E-Z Anesthesia, Braintree, MA, USA). Anesthesia was used exclusively during filament insertion and removal, whereas between these procedures the rats were taken out of anesthesia. The rats were decapitated at 24 h after tMCAO. The SO rats were subjected to a similar surgical procedure under anesthesia (neck incision and separation of the bifurcation), but without tMCAO. The respiratory rate and body temperature of the animal were monitored. These physiological parameters did not exceed the permissible values for each rat (body temperature: 37–38 °C; respiratory rate: not less than 33 per min). All rats were alive before decapitation.

Each experimental group consisted of at least five animals.

### 4.3. Magnetic Resonance Imaging (MRI)

The MRI study of the characteristics of the ischemic injury of rat brains was carried out using a small animal 7T ClinScan tomograph (Bruker BioSpin, Billerica, MA, USA) as previously described [31]. The control of blood flow was conducted as previously described [53].

### 4.4. Rat Brain Tissues

In rats that were subjected to tMCAO, the subcortical structures of the brain were isolated from the IH (“IR-i” group of samples) and the CH (“IR-c” group of samples). From the brain of SO rats, subcortical regions were isolated from the right hemisphere, corresponding to the IH of ischemic rats (“SO-r” group of samples) and the left hemisphere, corresponding to the CH of ischemic rats (“SO-l” group of samples). All IR-i, IR-c, SO-r, and SO-l samples were placed in RNAlater (Ambion, Austin, TX, USA) solution for 24 h at 4 °C and then stored at −70 °C.

### 4.5. RNA Isolation

Total RNA was isolated as previously described [40]. RNA integrity was checked using capillary electrophoresis (Experion, BioRad, Hercules, CA, USA). RNA integrity number (RIN) was at least 9.0.

### 4.6. RNA-Seq

The RNA-Seq analysis of the polyA fraction of the total RNA was conducted with the participation of ZAO Genoanalytika, Moscow, Russia, using an Illumina HiSeq 1500 instrument (Illumina, San Diego, CA, USA) as previously described [40]. At least 10 million reads (1/50 nt) were generated.

### 4.7. Reverse Transcription Polymerase Chain Reaction (RT–PCR) in Real-Time

cDNA synthesis was conducted as previously described [40]. Oligo (dT)_18_ primers were used to analyse mRNA.

The polymerase chain reaction (PCR) was conducted using a StepOnePlus Real-Time PCR System (Applied Biosystems, Foster City, CA, USA) as previously described [40]. Primers specific to the genes studied were selected using OLIGO Primer Analysis Software version 6.31 and were synthesized by the Evrogen Joint Stock Company (Appendix A). Each cDNA sample was analyzed three times.

### 4.8. Statistical Analysis 

Each of the comparison groups (IR-i, IR-c, SO-r, and SO-l) included three animals (n = 3) for RNA-Seq experiments. All genes were annotated using Cuffdiff/Cufflinks, so multiple isoforms (their RefSeq ID) were merged. The annotation database was available at genome.ucsc.edu/cgi-bin/hgTables (accessed on 15 April 2021). The levels of mRNA expression were measured using the Cuffdiff program as previously described [30]. Only genes that exhibited changes in expression >1.5-fold and had a *p*-values (*t*-test) adjusted using the Benjamini–Hochberg procedure lower 0.05 (*Padj* < 0.05) were considered.

Two reference genes *Gapdh* and *Rpl3* were used to normalize the RT–PCR results [54]. Calculations were performed using Relative Expression Software Tool (REST) 2005 software (gene-quantification, Freising-Weihenstephan, Bavaria, Germany) [55] as previously described [40]. The efficiency values for all PCR reactions were in the range 1.86 to 2.02 (Appendix A). Five animals were included in each comparison group. When comparing data groups, statistically significant differences were considered with the probability *p* < 0.05. Additional calculations were performed using Microsoft Excel (Microsoft Office 2010, Microsoft, Redmond, WA, USA). 

Hierarchical cluster analysis of DEGs was performed using Heatmapper (Wishart Research Group, University of Alberta, Ottawa, ON, Canada) [56]. A volcano plot was constructed by Microsoft Excel (Microsoft Office 2010, Microsoft, Redmond, WA, USA). Violin plots were constructed by BoxPlotR: a web-tool for generation of box plots [57].

### 4.9. Functional Analysis and Network Construction

The Database for Annotation, Visualization and Integrated Discovery (DAVID) (2021 Update) [58] and The PANTHER (Protein ANalysis THrough Evolutionary Relationships) [59] resources were used to annotate the functions of the differentially expressed mRNAs (DEGs). When comparing data groups, statistically significant differences were considered with the probability *Padj* < 0.05. 

The Cytoscape 3.8.2 software (Institute for Systems Biology, Seattle, WA, USA) [60] was used to visualize the regulatory network. Additional calculations were performed using Microsoft Excel (Microsoft Office 2010, Microsoft, Redmond, WA, USA).

### 4.10. Availability of Data and Material

RNA-sequencing data have been deposited in the Sequence Read Archive database under accession code PRJNA803984 (SAMN25694602-SAMN25694613, http://www.ncbi.nlm.nih.gov/bioproject/803984 (accessed on 18 April 2022)) [61].

## 5. Conclusions

A comprehensive analysis of the transcriptome changes that occur in the IH after focal stroke using two commonly used controls was carried out in a rat tMCAO model. As a result, the sets of genetic responses were characterized as possibly being inherent to the use of CH as a control. Thus, we identified traces of a bilateral genetic response in the brain after focal stroke.

## Figures and Tables

**Figure 1 ijms-23-07308-f001:**
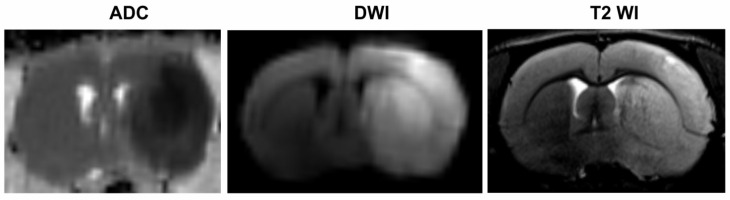
MRI of ischemic foci after tMCAO. DWI with an ADC map and T2 WI scans of the formation of ischemic injury areas with ipsilateral hemispheric (subcortex plus cortex) localization in the brain of rats at 24 h after tMCAO. Concomitantly, no pathological changes in the CH of ischemic rats observed according to the MRI data.

**Figure 2 ijms-23-07308-f002:**
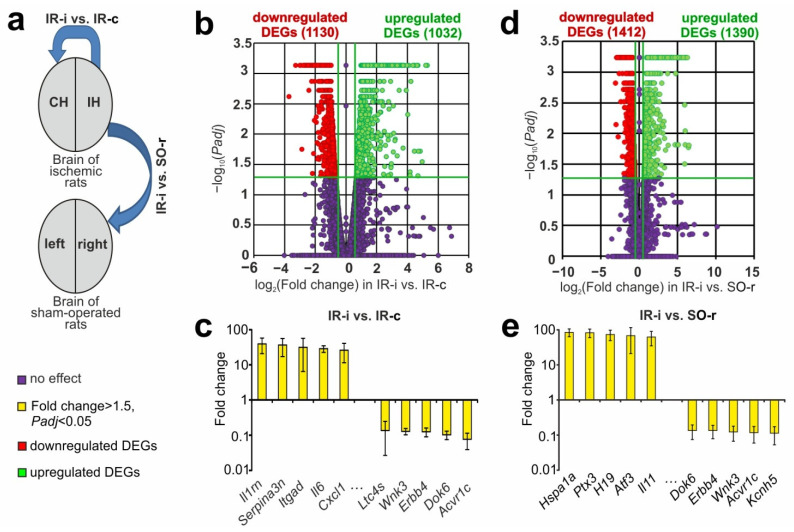
Genome-wide transcriptome changes recorded at 24 h after tMCAO in subcortical structures of the IH related to the sham-operated (SO) and contralateral controls. (**a**) The data comparison scheme is presented. The blue arrows indicate comparisons of IH samples from ischemic (IR) rats vs. right hemisphere samples of SO rats, as well as between IH and CH samples of IR animals. (**b**,**d**) The RNA-Seq results presented are for IR-i vs. IR-c (**b**) and IR-i vs. SO-r (**d**) using volcano plots. The cutoff for gene expression changes was a 1.50-fold change. Only those genes with *p*-values adjusted using the Benjamini–Hochberg procedure (*Padj* < 0.05) were selected for analysis. Up- and downregulated DEGs are represented as green and red dots, respectively. Genes that were not differentially expressed are presented as dark purple dots (fold change ≤ 1.50; *Padj* ≥ 0.05). (**c**,**e**) Top 10 genes that exhibited the greatest fold change in expression in IR-i vs. IR-c (**c**) and IR-i vs. SO-r (**e**). The data are presented as the mean ± standard error (SE) of the mean. Three animals were included in each comparison group.

**Figure 3 ijms-23-07308-f003:**
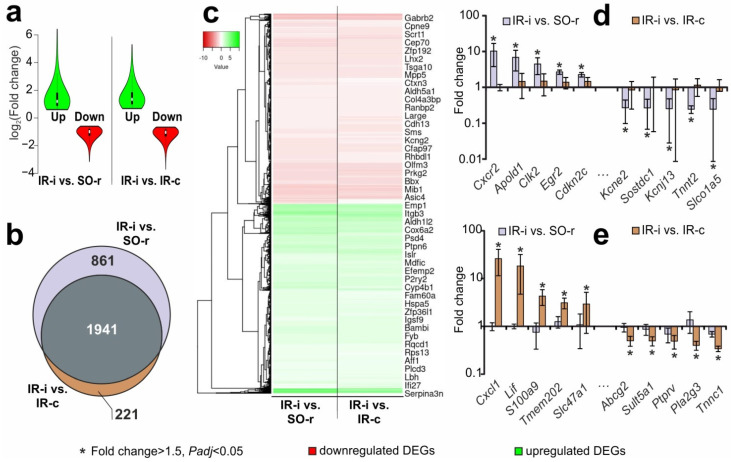
Comparisons of the results of DEGs in the subcortical structures of the IH at 24 h after tMCAO, obtained using CH and SO controls. (**a**) Violin plot comparing the distributions of the intensities of expression changes for DEGs from each pairwise comparison of IR-i vs. SO-r and IR-i vs. IR-c. (**b**) The Venn diagram represents the schematic comparisons of the results obtained using two controls in pairwise comparisons of IR-i vs. SO-r and IR-i vs. IR-c. (**c**) Hierarchical cluster analysis of all DEGs in IR-i vs. SO-r and IR-i vs. IR-c. Each column represents a comparison group, and each row represents a DEG. The green strips indicate high relative expression levels, and the red strips indicate low relative expression levels; n = 3 per group. (**d**) The top 10 genes that exhibited the greatest fold change in expression in IR-i vs. SO-r were non-DEGs that exhibited a fold change <1.50 in IR-i vs. IR-c. (**e**) The top 10 genes in IR-i vs. IR-c that were non-DEGs with a fold change <1.50 in IR-i vs. SO-r. The data are presented as the mean ± SE. Genes with a fold change >1.50 and *Padj* < 0.05 in the comparison group are marked with an asterisk (*).

**Figure 4 ijms-23-07308-f004:**
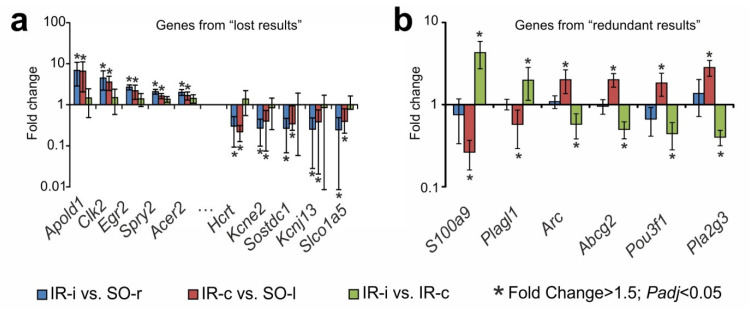
Genes that were DEGs in IR-c vs. SO-l and overlapped with sets of “lost” (**a**) and “redundant” (**b**) results, respectively. (**a**) The top 10 genes that exhibited the greatest fold change in expression in IR-c vs. SO-l were DEGs in IR-i vs. SO-r but non-DEGs that exhibited a fold change <1.50 in IR-i vs. IR-c. (**b**) The genes that were DEGs in IR-c vs. SO-l and IR-i vs. IR-c but non-DEGs in IR-i vs. SO-r. The data are presented as the mean ± SE.

**Figure 5 ijms-23-07308-f005:**
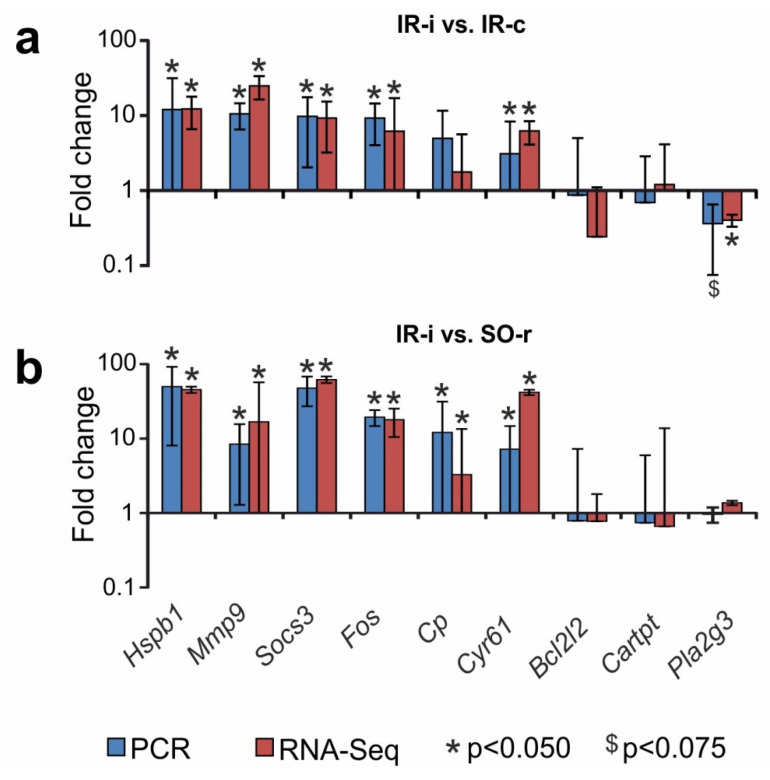
Real-time RT-PCR verification of the RNA-Seq results. Data for comparisons of IR-i vs. IR-c and IR-i vs. SO-r are shown. Nine (*Hspb1*, *Mmp9*, *Socs3*, *Fos*, *Cp*, *Cyr61*, *Bcl2l2*, *Cartpt*, and *Pla2g3*) genes were selected for PCR analysis. The reference *Gapdh* and *Rpl3* genes were used to normalize the PCR results. Five animals were included in each comparison group.

**Figure 6 ijms-23-07308-f006:**
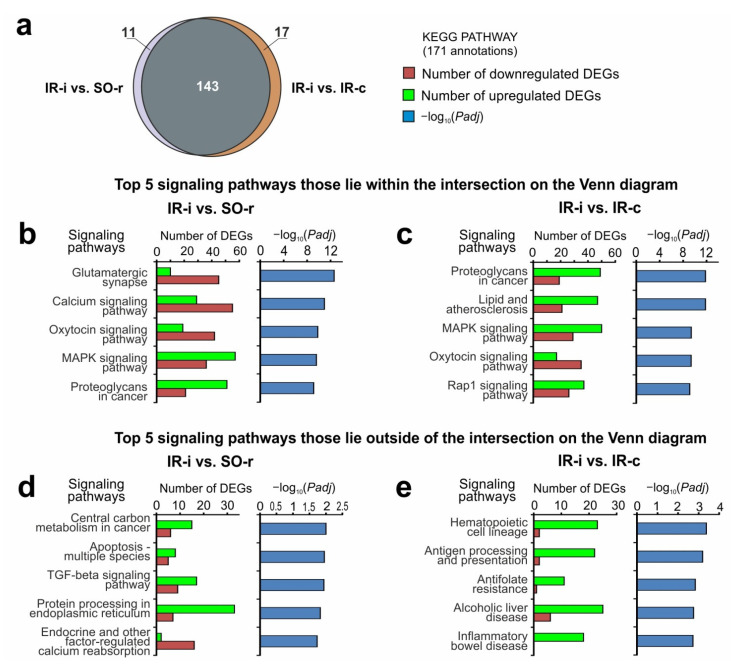
Search for common and unique signaling pathways associated with DEGs at 24 h after tMCAO in subcortical structures of the IH related to the sham-operated and contralateral controls. A pathway enrichment analysis of DEGs in two pairwise comparisons, i.e., IR-i vs. SO-r and IR-i vs. IR-c, was carried out according to DAVID (2021 Update). (**a**) Schematic comparison of the KEGG PATHWAY (KP) annotations associated with the DEGs obtained in two pairwise comparisons, i.e., IR-i vs. SO-r and IR-i vs. IR-c, as assessed using a Venn diagram. The numbers on the diagram segments indicate the number of annotations. (**b**,**c**) The most significant signaling pathways and corresponding *Padj*-values, as well as the number of up- and downregulated DEGs that lay within the intersection on the Venn diagram (**a**) in IR-i vs. SO-r and (**b**) in IR-i vs. IR-c (**c**). (**d**,**e**) The most significant signaling pathways and corresponding *Padj*-values, as well as the number of up- and downregulated DEGs that lay outside of the intersection on the Venn diagrams (**a**) in IR-i vs. SO-r and (**d**) in IR-i vs. IR-c (**e**). DEGs and signaling pathways with *Padj* < 0.05 were exclusively selected for the analysis. Three animals were included in each comparison group.

## Data Availability

Publicly available datasets were analyzed in this study. These data can be found here: Refs. [61,62].

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
