# Peer review of "Comparative Use of Contralateral and Sham-Operated Controls Reveals Traces of a Bilateral Genetic Response in the Rat Brain after Focal Stroke"

_ijms, 2022, doi:10.3390/ijms23137308_

Round 1

Reviewer 1 Report

The study is very interesting and well written. It is carried out by performing MRI analysis for the visualization of ischemic damage and genome-wide RNA sequencing for assessment of gene expression variations induced in response to focal cerebral ischemia in rats. Very intersting is the characterization of non-overlapping differentially expressed genes (DEGs) caming from transcriptome analysis carried out as comparison between the ipsilateral ischemic hemisphere (IH) vs. Sham-operated ipsilateral hemisphere (SO), or between the IH vs. the contralateral hemisphere (CH). The conclusion of the work suggests caution in using the CH as a control of the ischemic event which, as it is known, induces important transhemispheric effects. Therefore, the use of the CH side as technical control may prevent the identification of genes whose expression varies co-directionally and of the same entity on both sides.

Experiments are well designed. However, it should be helpful to add:

1-     A western blot expression analysis of selected overlapping and non-overlapping DEGs in the ischemic tisssue in comparison to SO and CH;

2-     A trascriptome analysis carried out as comparison between the Contralateral ischemic hemisphere (CH) vs. the sham-operated Contralateral hemisphere (SO-CH).

Moreover, it would be better to eliminate from the Conclusion paragraph (pag 12, lines 455-457) the following sentence: “Moreover, the specific transcriptome response of the distant brain areas should be considered when using it as a control in studies of target brain regions in other complex diseases with focal damage”. This sentence is too speculative and cannot represent a conclusion of this work of comparative transcriptome analysis carried out by using as control of ischemic brain the CH or SO, but not adjacent non-ischemic areas of the same hemisphere.

Author Response

Response to the comments of Reviewer 1 to Manuscript ID: ijms-1755974

Authors:

We are very grateful to the Reviewer 1 for the review and constructive comments. We carefully considered the comments of the Reviewer 1 and attached the answers to all comments.

Reviewer 1:

Experiments are well designed. However, it should be helpful to add:

  • A western blot expression analysis of selected overlapping and non-overlapping DEGs in the ischemic tisssue in comparison to SO and CH;

Authors:

We are very grateful to the Reviewer for the comment. However, the Western blot expression analysis is an independent study and plan to be performed by us in the future. Here, we consider the study of brain protein profile after ischemia as one of the limitations of our study. In accordance with the Reviewer’s recommendation, changes were added in the text (p. 10, line 347 in Mark-up copy).

Reviewer 1:                                                                                       

  • A trascriptome analysis carried out as comparison between the Contralateral ischemic hemisphere (CH) vs. the sham-operated Contralateral hemisphere (SO-CH).

Authors:

We are very grateful to the Reviewer for the comment. Indeed, we carried out a comparison between the Contralateral ischemic hemisphere (CH) vs. the sham-operated Contralateral hemisphere (SO-CH). As a result, we revealed 164 differentially expressed genes (DEGs) with cut-off>1.5 and Padj<0.05 that were associated with subcortical structures of CH at 24h after tMCAO. Among them, were DEGs for CH but non-DEGs for IH. Also, a number of genes had co-directed changes in expression in both hemispheres. Additionally, there were genes that had oppositely changed mRNA level in two brain hemispheres after tMCAO. Moreover, each of the gene sets had interesting functional annotations. Taking into account all of the above, in preparing this paper (ijms-1755974), we realized that the inclusion of results obtained using “SO-CH” strongly overburdens this paper. So, such results require independent discussion and a separate publication. Thus, we plan to publish the relevant paper about “SO-CH” results in the very near future.

Reviewer 1:

Moreover, it would be better to eliminate from the Conclusion paragraph (pag 12, lines 455-457) the following sentence: “Moreover, the specific transcriptome response of the distant brain areas should be considered when using it as a control in studies of target brain regions in other complex diseases with focal damage”. This sentence is too speculative and cannot represent a conclusion of this work of comparative transcriptome analysis carried out by using as control of ischemic brain the CH or SO, but not adjacent non-ischemic areas of the same hemisphere.

Authors:

In accordance with the Reviewer’s recommendation, changes were added in the text (p. 13, lines 488-492 in Mark-up copy).

Reviewer 2 Report

The authors wrote an excellent and clearly presented paper entitled "Comparative use of contralateral and sham-operated controls reveals traces of a bilateral genetic response in the rat brain after focal stroke". I have no doubt that this paper will be highly cited and make a significant positive impact on the field. I have only a minor comment for this paper: In the Material and Methods section, the authors need to describe how they analyzed genes with multiple isoforms in order to generate a final differentially expressed gene list.

Author Response

Response to the comments of Reviewer 2 to Manuscript ID: ijms-1755974

Authors:

We are very grateful to the Reviewer 2 for the review and constructive comments. We carefully considered the comments of the Reviewer 2 and attached the answers to all comments.

Reviewer 2:

The authors wrote an excellent and clearly presented paper entitled "Comparative use of contralateral and sham-operated controls reveals traces of a bilateral genetic response in the rat brain after focal stroke". I have no doubt that this paper will be highly cited and make a significant positive impact on the field. I have only a minor comment for this paper: In the Material and Methods section, the authors need to describe how they analyzed genes with multiple isoforms in order to generate a final differentially expressed gene list.

Authors:

The annotation database (RefSeq, rn5, genome.ucsc.edu/cgi-bin/hgTables) was used for differential expression analysis. Then annotation was processed using cufflinks/cuffdiff tool, so multiple isoforms (their RefSeq ID) have been merged. Therefore, we did not evaluate the contribution of each of the isoforms, but taking into account all isoforms, we characterized the differential expression of the gene encoding all its isoforms. In accordance with the Reviewer’s recommendation, changes were added in the text of Materials and Methods section (p. 12, lines 445-447 in Mark-up copy).

Reviewer 3 Report

The authors of this manuscript attempt to compare the gene expression between CH or SO with the tMCAO-challenged brains. The topic of this paper is relevant, timely, and of interest to the audience of this journal. Overall, the manuscript looks very interesting and the whole study seems to be acceptable. However, I have some concerns which need to be addressed.

1.      In figure 1, Please change the good quality MRI images. It is very hard to distinguish brain areas.

2.      The authors did the transient mouse MCAO model, but there is no mention of physiological parameter monitoring during/after surgery eg: BP, blood gases, pulse, heart rate, temperature, use of analgesics or blood flow monitoring during ischemia/reperfusion eg: Laser Doppler Blood Flow monitoring.

3.      Please discuss the rationale of this study in terms of clinical use.

4.      Mention any exclusion or inclusion criteria in this study.

5.      In the method section, there is insufficient technical information regarding the tMCAO model procedure.

6.      In some figure legend, missing mice number information (eg. Figure 4). Please confirm others also.

7.  Correct subheading number in “Results” and “Materials Methods” sections.

8. Add “Statistical Analysis” subheading in the Materials Methods section.

Author Response

Response to the comments of Reviewer 3 to Manuscript ID: ijms-1755974

Authors:

We are very grateful to the Reviewer 3 for the review and constructive comments. We carefully considered the comments of the Reviewer 3 and attached the answers to all comments.

Reviewer 3:

The authors of this manuscript attempt to compare the gene expression between CH or SO with the tMCAO-challenged brains. The topic of this paper is relevant, timely, and of interest to the audience of this journal. Overall, the manuscript looks very interesting and the whole study seems to be acceptable. However, I have some concerns which need to be addressed.

  • In figure 1, Please change the good quality MRI images. It is very hard to distinguish brain areas.

Authors:

In accordance with Reviewer’s recommendation, we have increased the resolution of ADC and DWI to 600 dpi (p. 3, lines 107-108 in Mark-up copy). Moreover, the raw images were downloaded to the IJMS submission system too. It should be noted that image resolution corresponds to the selected pulse sequences and the tomograph type. DWI and ADC are functional and difficult to obtain. Therefore their resolution is much worse than the anatomical image of T2 WI. MRI images of the same quality have been published by other researchers (Schäbitz et al. Stroke. 2004. 35(5):1175-9; Villa et al. Mol Med. 2007;13(3-4):125-33; Liu et al. BMC Neurosci 2012;13:154; Canazza et al. Front Neurol. 2014. 5:19).

Reviewer 3:

  • The authors did the transient mouse MCAO model, but there is no mention of physiological parameter monitoring during/after surgery eg: BP, blood gases, pulse, heart rate, temperature, use of analgesics or blood flow monitoring during ischemia/reperfusion eg: Laser Doppler Blood Flow monitoring.

Authors:

In accordance with the Reviewer’s recommendation, changes were added in the text of Materials and Methods section. The respiratory rate and body temperature of the animal were monitored. These physiological parameters did not exceed the permissible values for each rat (body temperature: 37-38 °Ð¡; respiratory rate: not less 33 per min) (p. 11, lines 377-380 in Mark-up copy). The control of blood flow was conducted using MRI as previously described (Gubskiy et al., 2018) (p. 11, lines 385-386 in Mark-up copy). Moreover, three-dimensional time-of-flight magnetic resonance angiography (3D-TOF MRA) was used for visualization of the main arteries and control of the recanalization as previously described (Filippenkov et al., 2021b). Unfortunately, the analysis of additional parameters was not provided for in the protocol for setting up the tMCAO. So, such manipulations, including use of analgesics, could cause a significant impact on the transcriptomic changes in rat brain cells  and may generate unacceptable noise in our research.

Reviewer 3:

  • Please discuss the rationale of this study in terms of clinical use.

Authors:

One of the problems for clinical trials is related to the correlation between the preclinical results obtained using animal models and human parameters (Fluri et al., 2015; Atkins et al., 2020; Narayan et al., 2021). It cannot be ruled out that the quality of clinical uses is limited by problem of control samples at the preclinical stage. Here, we characterize genes that can produce “redundant” or “lost” results of differential expression when using the contralateral hemisphere as a control under tMCAO conditions. We believe that the correct choice of controls will contribute to a more reliable interpretation of the data obtained using animals models of experimental ischemia at the preclinical stage. Thus, it is possible to achieve more effective diagnostic and therapeutic approaches to the treatment of ischemic stroke at the clinical stage. In accordance with the Reviewer’s recommendation, changes were added in the text of Discussion section (p. 10, lines 352-361 in Mark-up copy).

Reviewer 3:

  • Mention any exclusion or inclusion criteria in this study.

Authors:

In accordance with the Reviewer’s recommendation, exclusion or inclusion criteria were added in the text. White 2-month-old male rats of the Wistar line (weight, 200–250 g) were used (p. 10, line 364 in Mark-up copy). Based on MRI data, the ischemic injury was localized in ipsilateral hemisphere (subcortex plus cortex) of the brain at 24 h after tMCAO in all rats from IR group (p. 3, lines 101-103 in Mark-up copy). All rats were alive before decapitation (p. 11, line 380 in Mark-up copy). Only such rats were used for transcriptome analysis.

Reviewer 3:

  • In the method section, there is insufficient technical information regarding the tMCAO model procedure.

Authors:

In accordance with the Reviewer’s recommendation, technical information regarding the tMCAO model procedure was added in the Supplementary Method S1 and in the text (p. 10, lines 370-371; p. 13, line 498 in Mark-up copy)

Reviewer 3:

  • In some figure legend, missing mice number information (eg. Figure 4). Please confirm others also.

Authors:

In accordance with the Reviewer’s recommendation, changes were added in the legend of Figures (p. 4, lines 138-139; p. 5, line 182; p. 6, line 203; p. 8, line 261 in Mark-up copy)

Reviewer 3:

  • Correct subheading number in “Results” and “Materials Methods” sections.

Authors:

In accordance with the Reviewer’s recommendation, changes were added in the text (lines 95, 113, 140, 151, 187, 204, 363, 367, 382, 387, 394, 404, 424, 442, 468, 481 in Mark-up copy).

Reviewer 3:

  • Add “Statistical Analysis” subheading in the Materials Methods section.

Authors:

In accordance with the Reviewer’s recommendation, changes were added in the text (p. 12, lines 442-467 in Mark-up copy).

Round 2

Reviewer 1 Report

The authors failed to respond to several requests raised for the work revision. As requested, a western blot expression analysis of at least some selected overlapping and non-overlapping DEGs and the trascriptome analysis performed as comparison between the contralateral ischemic hemisphere (CH) vs. the sham-operated contralateral hemisphere (SO-CH) should be included in this work. The authors did not add this data in the revised version of the manuscript and are planning to publish these data later. However, as required, such data should be added to this work. For these reasons, the work in its current form is not adeguate for publication.

Author Response

Response to the comments of Reviewer 1 to Manuscript ID: ijms-1755974

Authors:

We are very grateful to the Reviewer 1 for the review and constructive comments. We carefully considered the comments of the Reviewer 1 and attached the answers to all comments.

Reviewer 1:

The authors failed to respond to several requests raised for the work revision. As requested, a western blot expression analysis of at least some selected overlapping and non-overlapping DEGs and the trascriptome analysis performed as comparison between the contralateral ischemic hemisphere (CH) vs. the sham-operated contralateral hemisphere (SO-CH) should be included in this work. The authors did not add this data in the revised version of the manuscript and are planning to publish these data later. However, as required, such data should be added to this work. For these reasons, the work in its current form is not adeguate for publication.

Authors:

In accordance with the Reviewer’s recommendation, the transcriptome analysis performed as comparison between the contralateral ischemic hemisphere vs. the sham-operated contralateral hemisphere was included in this work. The changes were added in Figure 4, Supplementary Figure S1 and in the text (p. 6, lines 185-207; p. 10, lines 333-335; p. 11, lines 419-421; p. 12, lines 441-443; p. 13, lines 480-482 in Mark-up copy).

We disagree with the Reviewer's requirement to include additional studies of Western blot analysis. Well known, the correlation between transcriptome and proteome profiles is non-linear because of post-transcriptional, translational, and post-translational regulatory events (Gry et al., 2009; Franks et al., 2017; Eraslan et al., 2019; Buccitelli and Selbach, 2020). Therefore, the transcriptome profile does not provide reliable criteria for choosing individual proteins for Western analysis to show the difference in the use of the SO and CH controls at the protein level. Thus, Western blot expression analysis cannot be among of objectives of this study. The changes were added in the text (p. 10, lines 370-375 in Mark-up copy).

Reviewer 3 Report

I have no further comments and suggestions for authors.

Author Response

Response to the comments of Reviewer 3 to Manuscript ID: ijms-1755974

Reviewer 3:

I have no further comments and suggestions for authors.

Authors:

We are very grateful to the Reviewer 3 for the review.

Round 3

Reviewer 1 Report

The addition of trascriptome analysis performed as comparison between the contralateral ischemic hemisphere vs. the sham-operated contralateral left hemisphere was fundamental and enriched the work. It would be useful to also provide a corresponding table of DEGs in the supplementary materials.

Even if it is well known that there is not a linear correlation between transcriptome and proteome profiles, the suggestion to provide a western blot analysis of selected DEGs was made because it could have offered more value to the study.

The work in its current form is now adeguate for publication.